# Computing Full Conformal Prediction Set with Approximate Homotopy

**Eugene Ndiaye**
RIKEN Center for Advanced Intelligence Project
`eugene.ndiaye@riken.jp`

**Ichiro Takeuchi**
Nagoya Institute of Technology
`takeuchi.ichiro@nitech.ac.jp`

## Abstract

If you are predicting the label $y$ of a new object with $\hat{y}$, how confident are you that $y = \hat{y}$? Conformal prediction methods provide an elegant framework for answering such question by building a $100(1 - \alpha)\%$ confidence region without assumptions on the distribution of the data. It is based on a refitting procedure that parses all the possibilities for $y$ to select the most likely ones. Although providing strong coverage guarantees, conformal set is impractical to compute exactly for many regression problems. We propose efficient algorithms to compute conformal prediction set using approximated solution of (convex) regularized empirical risk minimization. Our approaches rely on a new homotopy continuation technique for tracking the solution path with respect to sequential changes of the observations. We also provide a detailed analysis quantifying its complexity.

## 1  Introduction

In many practical applications of regression models it is beneficial to provide, not only a point-prediction, but also a prediction set that has some desired coverage property. This is especially true when a critical decision is being made based on the prediction, e.g., in medical diagnosis or experimental design. *Conformal prediction* is a general framework for constructing non-asymptotic and distribution-free prediction sets. Since the seminal work of [23, 21], the statistical properties and computational algorithms for conformal prediction have been developed for a variety of machine learning problems such as density estimation, clustering, and regression - see the review of [2].

Let $\mathcal{D}_n = \{(x_1, y_1), \cdots, (x_n, y_n)\}$ be a sequence of features and labels of random variables in $\mathbb{R}^p \times \mathbb{R}$ from a distribution $\mathbb{P}$. Based on observed data $\mathcal{D}_n$ and a new test instance $x_{n+1}$ in $\mathbb{R}^p$, the goal of conformal prediction is to build a $100(1 - \alpha)\%$ confidence set that contains the unobserved variable $y_{n+1}$ for $\alpha$ in $(0, 1)$, without any specific assumptions on the distribution $\mathbb{P}$.

The conformal prediction set for $y_{n+1}$ is defined as the set of $z \in \mathbb{R}$ whose *typicalness* is sufficiently large. The typicalness of each $z$ is defined based on the residuals of the regression model, trained with an augmented training set $\mathcal{D}_{n+1}(z) = \mathcal{D}_n \cup (x_{n+1}, z)$. On average, prediction sets constructed within a conformal prediction framework are shown to have a desirable coverage property, as long as the training instances $\{(x_i, y_i)\}_{i=1}^{n+1}$ are exchangeable, and the regression estimator is symmetric with respect to the training instances (even when the model is not correctly specified).

Despite these attractive properties, the computation of conformal prediction sets has been intractable since one needs to fit infinitely many regression models with an augmented training set $\mathcal{D}_{n+1}(z)$, for all possible $z \in \mathbb{R}$. Except for simple regression estimators with quadratic loss (such as least-square regression, ridge regression or lasso estimators) where an explicit and exact solution of the model parameter can be written as a piece of a linear function in the observation vectors, the computation of the full and exact conformal set for the general regression problem is challenging and still open.

**Contributions.** We propose a general method to compute the full conformal prediction set for a wider class of regression estimators. The main novelties are summarized in the following points:

- We introduce a new homotopy continuation technique, inspired by [8, 16], which can efficiently update an approximate solution with tolerance $\epsilon > 0$, when the data are streamed sequentially. For this, we show that the variation of the optimization error only depends on the loss on the new input data. Thus, exploiting the regularity of the loss, we can provide a range of observations for which an approximate solution is still valid. This allows us to approximately fit infinitely many regression models for all possible $z$ in a pre-selected range $[y_{\min}, y_{\max}]$, using only a finite number of candidate $z$. For example, when the loss function is smooth, the number of model fittings required for constructing the prediction set is $O(1/\sqrt{\epsilon})$.
- Exploiting the approximation error bounds of the proposed homotopy continuation method, we can construct the prediction set based on the $\epsilon$-solution, which satisfies the same valid coverage properties under the same mild assumptions as the conformal prediction framework. When the approximation tolerance $\epsilon$ decreases to $0$, the prediction set converges to the *exact* conformal prediction set which would be obtained by fitting an infinitely large number of regression models. Furthermore, if the loss function of the regression estimator is smooth and some other regularity conditions are satisfied, the prediction set constructed by the proposed method is shown to contain the *exact* conformal prediction set.

For reproducibility, our implementation is available in

```
https://github.com/EugeneNdiaye/homotopy_conformal_prediction
```

**Notation.** For a non zero integer $n$, we denote $[n]$ to be the set $\{1, \cdots, n\}$. The dataset of size $n$ is denoted $\mathcal{D}_n = (x_i, y_i)_{i \in [n]}$, the row-wise feature matrix $X = [x_1, \cdots, x_{n+1}]^\top$, and $X_{[n]}$ is its restriction to the $n$ first rows. Given a proper, closed and convex function $f : \mathbb{R}^n \to \mathbb{R} \cup \{+\infty\}$, we denote $\mathrm{dom} f = \{x \in \mathbb{R}^n : f(x) < +\infty\}$. Its Fenchel-Legendre transform is $f^* : \mathbb{R}^n \to \mathbb{R} \cup \{+\infty\}$ defined by $f^*(x^*) = \sup_{x \in \mathrm{dom} f} \langle x^*, x \rangle - f(x)$. The smallest integer larger than a real value $r$ is denoted $\lceil r \rceil$. We denote by $Q_{1-\alpha}$, the $(1-\alpha)$-quantile of a real valued sequence $(U_i)_{i \in [n+1]}$, defined as the variable $Q_{1-\alpha} = U_{(\lceil (n+1)(1-\alpha) \rceil)}$, where $U_{(i)}$ are the $i$-th order statistics. For $j$ in $[n+1]$, the rank of $U_j$ among $U_1, \cdots, U_{n+1}$ is defined as $\mathrm{Rank}(U_j) = \sum_{i=1}^{n+1} \mathbb{1}_{U_i \le U_j}$. The interval $[a - \tau, a + \tau]$ will be denoted $[a \pm \tau]$.

## 2 Background and Problem Setup

We consider the framework of regularized empirical risk minimization (see for instance [22]) with a convex loss function $\ell : \mathbb{R} \times \mathbb{R} \mapsto \mathbb{R}$, a convex regularizer $\Omega : \mathbb{R} \mapsto \mathbb{R}$ and a positive scalar $\lambda$:

$$\hat{\beta} \in \arg\min_{\beta \in \mathbb{R}^p} P(\beta) := \sum_{i=1}^{n} \ell(y_i, x_i^\top \beta) + \lambda \Omega(\beta) \ . \tag{1}$$

For simplicity, we will assume that for any real values $z$ and $z_0$, we have $\ell(z_0, z)$ and $\ell(z, z_0)$ are non negative, $\ell(z_0, z_0)$ and $\ell^*(z_0, 0)$ are equal to zero. These assumptions are easy to satisfy and we refer the reader to the appendix for more details.

**Examples.** A popular example of a loss function found in the literature is `power norm regression`, where $\ell(a, b) = |a - b|^q$. When $q = 2$, this corresponds to classical linear regression. Cases where $q \in [1, 2)$ are common in robust statistics. In particular, $q = 1$ is known as least absolute deviation. The `logcosh` loss $\ell(a, b) = \gamma \log(\cosh(a - b)/\gamma)$ is a differentiable alternative to the $\ell_\infty$-norm (Chebychev approximation). One can also have the `Linex` loss function [9, 4] which provides an asymmetric loss $\ell(a, b) = \exp(\gamma(a - b)) - \gamma(a - b) - 1$, for $\gamma \neq 0$. Any convex regularization functions $\Omega$ *e.g.* Ridge [10] or sparsity inducing norms in [1] can be considered.

For a new test instance $x_{n+1}$, the goal is to construct a prediction set $\hat{\Gamma}^{(\alpha)}(x_{n+1})$ for $y_{n+1}$ such that

$$\mathbb{P}^{n+1}(y_{n+1} \in \hat{\Gamma}^{(\alpha)}(x_{n+1})) \ge 1 - \alpha \ \text{ for } \alpha \in (0, 1) \ . \tag{2}$$

## 2.1 Conformal Prediction

Conformal prediction [23] is a general framework for constructing confidence sets, with the remarkable properties of being distribution free, having a finite sample coverage guarantee, and being able to be adapted to any estimator under mild assumptions. We recall the arguments in [21, 14].

Let us introduce the extension of the optimization problem (1) with augmented training data $\mathcal{D}_{n+1}(z) := \mathcal{D}_n \cup \{(x_{n+1}, z)\}$ for $z \in \mathbb{R}$:

$$\hat{\beta}(z) \in \underset{\beta \in \mathbb{R}^p}{\arg\min} \, P_z(\beta) := \sum_{i=1}^{n} \ell(y_i, x_i^\top \beta) + \ell(z, x_{n+1}^\top \beta) + \lambda \Omega(\beta) \ . \tag{3}$$

Then, for any $z$ in $\mathbb{R}$, we define the conformity measure for $\mathcal{D}_{n+1}(z)$ as

$$\forall i \in [n], \ \hat{R}_i(z) = \psi(y_i, x_i^\top \hat{\beta}(z)) \text{ and } \hat{R}_{n+1}(z) = \psi(z, x_{n+1}^\top \hat{\beta}(z)) \ , \tag{4}$$

where $\psi$ is a real-valued function that is invariant with respect to any permutation of the input data. For example, in a linear regression problem, one can take the absolute value of the residual to be a conformity measure function *i.e.* $\hat{R}_i(z) = |y_i - x_i^\top \hat{\beta}(z)|$.

The main idea for constructing a conformal confidence set is to consider the *typicalness* of a candidate point $z$ measured as

$$\hat{\pi}(z) = \hat{\pi}(\mathcal{D}_{n+1}(z)) := 1 - \frac{1}{n+1} \text{Rank}(\hat{R}_{n+1}(z)) \ . \tag{5}$$

If the sequence $(x_i, y_i)_{i \in [n+1]}$ is exchangeable and identically distributed, then $(\hat{R}_i(y_{n+1}))_{i \in [n+1]}$ is also , by the invariance of $\hat{R}$ *w.r.t.* permutations of the data. Since the rank of one variable among an exchangeable and identically distributed sequence is (sub)-uniformly distributed (see [3]) in $\{1, \cdots, n+1\}$, we have $\mathbb{P}^{n+1}(\hat{\pi}(y_{n+1}) \leq \alpha) \leq \alpha$ for any $\alpha$ in $(0, 1)$. This implies that the function $\hat{\pi}$ takes a small value on atypical data. Classical statistics for hypothesis testing, such as a $p$-value function, satisfy such a condition under the null hypothesis (see [12, Lemma 3.3.1]). In particular, this implies that the desired coverage guarantee in Equation (2) is verified by the conformal set defined as

$$\hat{\Gamma}^{(\alpha)}(x_{n+1}) := \{z \in \mathbb{R} : \hat{\pi}(z) > \alpha\} \ . \tag{6}$$

The conformal set gathers the real value $z$ such that $\hat{\pi}(z) > \alpha$, if and only if $\hat{R}_{n+1}(z)$ is ranked no higher than $\lceil (n+1)(1-\alpha) \rceil$, among $\hat{R}_i(z)$ for all $i$ in $[n]$. For regression problems where $y_{n+1}$ lies in a subset of $\mathbb{R}$, obtaining the conformal set $\hat{\Gamma}^{(\alpha)}(x_{n+1})$ in Equation (6) is computationally challenging. It requires re-fitting the prediction model $\hat{\beta}(z)$ for infinitely many candidates $z$ in $\mathbb{R}$ in order to compute a conformity measure such as $\hat{R}_i(z) = |y_i - x_i^\top \hat{\beta}(z)|$.

**Existing Approaches for Computing a Conformal Prediction Set.** In Ridge regression, for any $x$ in $\mathbb{R}^p$, $z \mapsto x^\top \hat{\beta}(z)$ is a linear function of $z$, implying that $\hat{R}_i(z)$ is piecewise linear. Exploiting this fact, an exact conformal set $\hat{\Gamma}^{(\alpha)}(x_{n+1})$ for Ridge regression was efficiently constructed in [18]. Similarly, using the piecewise linearity in $z$ of the Lasso solution, [13] proposed a piecewise linear homotopy under mild assumptions, when a single input sample point is perturbed. Apart from these cases of quadratic loss with Ridge and Lasso regularization, where an explicit formula of the estimator is available, computing such a set is often infeasible. Also, a known drawback of exact path computation is its exponential complexity in the worst case [7], and numerical instabilities due to multiple inversions of potentially ill-conditioned matrices.

Another approach is to split the dataset into a training set - in which the regression model is fitted, and a calibration set - in which the conformity scores and their ranks are computed. Although this approach avoids the computational bottleneck of the full conformal prediction framework, statistical efficiencies are lost both in the model fitting stage and in the conformity score rank computation stage, due to the effect of a reduced sample size. It also adds another layer of randomness, which may be undesirable for the construction of prediction intervals [13].

A common heuristic approach in the literature is to evaluate the typicalness $\hat{\pi}(z)$ only for an arbitrary finite number of grid points. Although the prediction set constructed by those finite number of $\hat{\pi}(z)$ might roughly mimic the conformal prediction set, the desirable coverage properties are no longer maintained. To overcome this issue, [5] proposed a discretization strategy with a more careful procedure to round the observation vectors, but failed to exactly preserve the $1-\alpha$ coverage guarantee. In the appendix, we discuss in detail critical limitations of such an approach.

**Algorithm 1** $\epsilon$-`online_homotopy`

---

**Input:** $\mathcal{D}_n = \{(x_1, y_1), \cdots, (x_n, y_n)\}, x_{n+1}, [y_{\min}, y_{\max}], \epsilon_0 < \epsilon$

Initialization: $z_{t_0} = x_{n+1}^\top \beta$ where $\beta$ is an $\epsilon_0$-solution for the problem (1) using only $\mathcal{D}_n$

**repeat**

    $z_{t_{k+1}} = z_{t_k} \pm s_\epsilon$ where $s_\epsilon = \sqrt{\frac{2}{\nu}(\epsilon - \epsilon_0)}$ if the loss is $\nu$-smooth

    Get $\beta(z_{t_{k+1}})$ by minimizing $P_{z_{t_{k+1}}}$ up to accuracy $\epsilon_0$ {warm started with $\beta(z_{t_k})$}

**until** $[y_{\min}, y_{\max}]$ is covered

**Return:** $\{z_{t_k}, \beta(z_{t_k})\}_{k \in [T_\epsilon]}$

---

## 3 Homotopy Algorithm

In constructing an exact conformal set, we need to be able to compute the entire path of the model parameters $\hat{\beta}(z)$; which is obtained after solving the augmented optimization problem in Equation (3), for any $z$ in $\mathbb{R}$. In fact, two problems arise. First, even for a single $z$, $\hat{\beta}(z)$ may not be available because, in general, the optimization problem *cannot* be solved exactly [17, Chapter 1]. Second, except for simple regression problems such as Ridge or Lasso, the entire exact path of $\hat{\beta}(z)$ cannot be computed infinitely many times.

Our basic idea to circumvent this difficulty is to rely on approximate solutions at a given precision $\epsilon > 0$. Here, we call an $\epsilon$-solution any vector $\beta$ such that its objective value satisfies

$$P_z(\beta) - P_z(\hat{\beta}(z)) \leq \epsilon \ . \tag{7}$$

An $\epsilon$-solution can be found efficiently, under mild assumptions on the regularity of the function being optimized. In this section, we show that finite paths of $\epsilon$-solutions can be computed for a wider class of regression problems. Indeed, it is not necessary to re-calculate a new solution for neighboring observations - *i.e.* $\beta(z)$ and $\beta(z_0)$ have the same performance when $z$ is close to $z_0$. We develop a precise analysis of this idea. Then, we show how this can be used to effectively approximate the conformal prediction set in Equation (6) based on exact solution, while preserving the coverage guarantee.

We recall the dual formulation [20, Chapter 31] of Equation (3):

$$\hat{\theta}(z) \in \underset{\theta \in \mathbb{R}^{n+1}}{\arg\max} D_z(\theta) := -\sum_{i=1}^{n} \ell^*(y_i, -\lambda\theta_i) - \ell^*(z, -\lambda\theta_{n+1}) - \lambda\Omega^*(X^\top \theta) \ . \tag{8}$$

For a primal/dual pair of vectors $(\beta(z), \theta(z))$ in $\mathrm{dom}P_z \times \mathrm{dom}D_z$, the duality gap is defined as

$$\mathrm{Gap}_z(\beta(z), \theta(z)) := P_z(\beta(z)) - D_z(\theta(z)) \ .$$

Weak duality ensures that $P_z(\beta(z)) \geq D_z(\theta(z))$, which yields an upper bound for the approximation error of $\beta(z)$ in Equation (7) *i.e.*

$$P_z(\beta(z)) - P_z(\hat{\beta}(z)) \leq \mathrm{Gap}_z(\beta(z), \theta(z)) \ .$$

This will allow us to keep track of the approximation error when the parameters of the objective function change. Given any $\beta$ such that $\mathrm{Gap}(\beta, \theta) \leq \epsilon$ *i.e.* an $\epsilon$-solution for problem (1), we explore the candidates for $y_{n+1}$ with the parameterization of the real line $z_t$ defined as

$$z_t := z_0 + t, \text{ for } t \in \mathbb{R} \text{ and } z_0 = x_{n+1}^\top \beta \ . \tag{9}$$

This additive parameterization was used in [13] for the case of the Lasso. It provides the nice property that adding $(x_{n+1}, z_0)$ as the $(n+1)$-th observation does not change the objective value of $\beta$ *i.e.* $P(\beta) = P_{z_0}(\beta)$. Thus, if a vector $\beta$ is an $\epsilon$-solution for $P$, it will remain so for $P_{z_0}$. Interestingly, such a choice is still valid for a sufficiently small $t$. We show that, depending on the regularity of the loss function, we can precisely derive a range of the parameter $t$ so that $\beta$ remains a valid $\epsilon$-solution for $P_{z_t}$ when the dataset $\mathcal{D}_n$ is augmented with $\{(x_{n+1}, z_t)\}$.

We define the variation of the duality gap between real values $z$ and $z_0$ to be

$$\Delta G(x_{n+1}, z, z_0) := \mathrm{Gap}_z(\beta, \theta) - \mathrm{Gap}_{z_0}(\beta, \theta) \ .$$

**Lemma 1.** *For any $(\beta, \theta) \in \mathrm{dom}P_w \times \mathrm{dom}D_w$ for $w \in \{z_0, z\}$, we have*

$$\Delta G(x_{n+1}, z, z_0) = [\ell(z, x_{n+1}^\top \beta) - \ell(z_0, x_{n+1}^\top \beta)] + [\ell^*(z, -\lambda\theta_{n+1}) - \ell^*(z_0, -\lambda\theta_{n+1})] \ .$$

Lemma 1 showed that the variation of the duality gap between $z$ and $z_0$ depends only on the variation of the loss function $\ell$, and its conjugate $\ell^*$. Thus, it is enough to exploit the regularity (*e.g.* smoothness) of the loss function in order to obtain an upper bound for the variation of the duality gap (and therefore the optimization error).

**Construction of Dual Feasible Vector.** A generic method for producing a dual-feasible vector is to re-scale the output of the gradient mapping. For a real value $z$, let $\beta(z)$ be any primal vector and let us denote $Y_z = (y_1, \cdots, y_n, z)$.

Optimality conditions for (3) and (8) implies $\hat{\theta}(z) = -\nabla\ell(Y_z, X\hat{\beta}(z))/\lambda$, which suggests we can make use of [16]

$$\theta(z) := \frac{-\nabla\ell(Y_z, X\beta(z))}{\max\{\lambda_t, \sigma^\circ_{\mathrm{dom}\Omega^*}(X^\top\nabla\ell(Y_z, X\beta(z)))\}} \in \mathrm{dom}D_z \ , \tag{10}$$

where $\sigma$ is the support function and $\sigma^\circ$ its polar function. When the regularization is a norm $\Omega(\cdot) = \|\cdot\|$, then $\sigma^\circ_{\mathrm{dom}\Omega^*}$ is the associated dual norm $\|\cdot\|_*$. When $\Omega$ is strongly convex, then the dual vector in Equation (10) simplifies to $\theta(z) = -\nabla\ell(Y_z, X\beta(z))/\lambda$.

Using $\theta(z_0)$ in Equation (10) with $z_0 = x_{n+1}^\top \beta$ greatly simplifies the expression for the variation of the duality gap between $z_t$ and $z_0$ in Lemma 1 to

$$\Delta G(x_{n+1}, z_t, z_0) = \ell(z_t, x_{n+1}^\top \beta) \ .$$

This directly follows from the assumptions $\ell(z_0, z_0) = \ell^*(z_0, 0) = 0$ and by construction of the dual vector $\theta_{n+1} \propto \partial_2\ell(z_0, x_{n+1}^\top\beta) = \partial_2\ell(z_0, z_0) = 0$. Whence, assuming that the loss function is $\nu$-smooth (see the appendix for more details and extensions to other regularity assumptions) and using the parameterization in Equation (9), we obtain

$$\Delta G(x_{n+1}, z_t, z_0) \leq \frac{\nu}{2}(z_t - z_0)^2 = \frac{\nu}{2}t^2 \ .$$

**Proposition 1.** *Assuming that the loss function $\ell$ is $\nu$-smooth, the variations of the gap $\Delta G(x_{n+1}, z_t, z_0)$ are smaller than $\epsilon$ for all $t$ in $[-\sqrt{2\epsilon/\nu}, \sqrt{2\epsilon/\nu}]$. Moreover, assuming that $\mathrm{Gap}_{z_0}(\beta(z_0), \theta(z_0)) \leq \epsilon_0 < \epsilon$, we have $(\beta(z_0), \theta(z_0))$ being a primal/dual $\epsilon$-solution for the optimization problem (3) with augmented data $\mathcal{D}_n \cup \{(x_{n+1}, z_t)\}$ as long as*

$$|z_t - z_0| \leq \sqrt{\frac{2}{\nu}(\epsilon - \epsilon_0)} =: s_\epsilon \ .$$

**Complexity.** A given interval $[y_{\min}, y_{\max}]$ can be covered by Algorithm 1 with $T_\epsilon$ steps where

$$T_\epsilon \leq \left\lceil \frac{y_{\max} - y_{\min}}{s_\epsilon} \right\rceil \in O\left(\frac{1}{\sqrt{\epsilon}}\right) \ .$$

We can notice that the step sizes $s_\epsilon$ (smooth case) for computing the whole path are independent of the data and the intermediate solutions. Thus, for computational efficiency, the latter can be computed in parallel or by sequentially warm-starting the initialization. Also, since the grid can be constructed by decreasing or increasing the value of $z_t$, one can observe that the number of solutions calculated along the path can be halved by using only $\beta(z_t)$ as an $\epsilon$-solution on the whole interval $[z_t \pm s_\epsilon]$.

**Lower Bound.** Using the same reasoning when the loss is $\mu$-strongly convex, we have

$$\Delta G(x_{n+1}, z_t, z_0) \geq \frac{\mu}{2}(z_t - z_0)^2 \ .$$

Hence $\Delta G(x_{n+1}, z_t, z_0) > \epsilon$ as soon as $|z_t - z_0| > \sqrt{\frac{2}{\mu}(\epsilon - \epsilon_0)}$. Thus, in order to guarantee $\epsilon$ approximation errors at any candidate $z_t$, all the step sizes are necessarily of order $\sqrt{\epsilon}$.

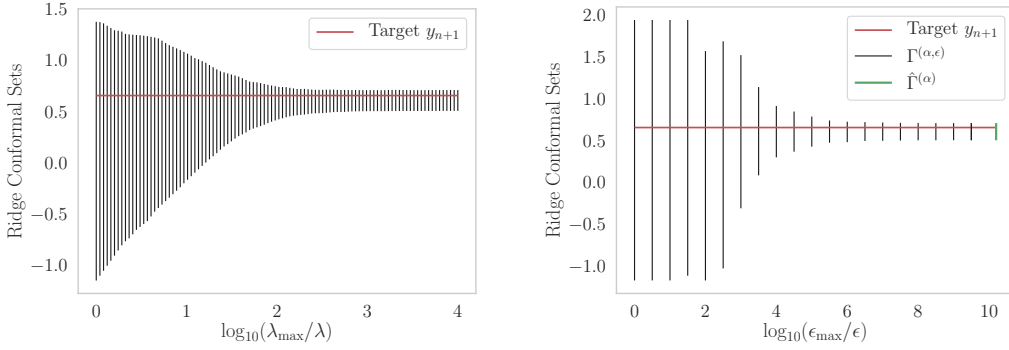

(a) Exact conformal prediction set for ridge regression with one hundred regularization parameters ranging from $\lambda_{\max} = \log(p)$ to $\lambda_{\min} = \lambda_{\max}/10^4$, spaced evenly on a log scale.

(b) Evolution of the conformal set of the proposed homotopy method with different optimization errors, spaced evenly on a geometric scale ranging from $\epsilon_{\max} = \|(y_1, \cdots, y_n)\|^2$ to $\epsilon_{\min} = \epsilon_{\max}/10^{10}$.

Figure 1: Illustration of conformal prediction sets at level $\alpha = 0.1$ with exact solutions and approximate solutions for ridge regression. We use a synthetic data set generated using `sklearn` with $X, y = $ `make_regression`$(n = 100, p = 50)$. We have chosen the hyperparameter with the smallest confidence set in Figure (a) to generate Figure (b).

**Choice of** $[y_{\min}, y_{max}]$. We follow the actual practice in the literature [13, Remark 5] and set $y_{\min} = y_{(1)}$ and $y_{\max} = y_{(n)}$. In that case, we have $\mathbb{P}(y_{n+1} \in [y_{\min}, y_{\max}]) \geq 1 - 2/(n+1)$. This implies a loss in the coverage guarantee of $2/(n+1)$, which is negligible when $n$ is sufficiently large.

**Related Works on Approximate Homotopy.** Recent papers [8, 16] have developed approximation path methods when a function is concavely parameterized. Such techniques cannot be used here since, for any $\beta \in \mathbb{R}^p$, the function $z \mapsto P_z(\beta)$ is not concave. Thus, it does not fit within their problem description.

Using homotopy continuation to update an exact Lasso solution in the online setting was performed by [6, 13]. Allowing an approximate solution allows us to extensively generalize those approaches to a broader class of machine learning tasks, with a variety of regularity assumptions.

## 4 Practical Computation of a Conformal Prediction Set

We present how to compute a conformal prediction set, based on the approximate homotopy algorithm in Section 3. We show that the set obtained preserves the coverage guarantee, and tends to the exact set when the optimization error $\epsilon$ decreases to zero. In the case of a smooth loss function, we present a variant of conformal sets with an approximate solution, which contains the exact conformal set.

### 4.1 Conformal Sets Directly Based on Approximate Solution

For a real value $z$, we cannot evaluate $\hat{\pi}(z)$ in Equation (5) in many cases because it depends on the exact solution $\hat{\beta}(z)$, which is unknown. Instead, we only have access to a given $\epsilon$-solution $\beta(z)$ and the corresponding (approximate) conformity measure given as:

$$\forall i \in [n], \ R_i(z) = \psi(y_i, x_i^\top \beta(z)) \text{ and } R_{n+1}(z) = \psi(z, x_{n+1}^\top \beta(z)) \ . \tag{11}$$

However, for establishing a coverage guarantee, one can note that *any* estimator that preserves exchangeability can be used. Whence, we define

$$\pi(z, \epsilon) := 1 - \frac{1}{n+1}\text{Rank}(R_{n+1}(z)), \qquad \Gamma^{(\alpha,\epsilon)}(x_{n+1}) := \{z \in \mathbb{R} : \pi(z, \epsilon) > \alpha\} \ . \tag{12}$$

**Proposition 2.** *Given a significance level $\alpha \in (0,1)$ and an optimization tolerance $\epsilon > 0$, if the observations $(x_i, y_i)_{i \in [n+1]}$ are exchangeable and identically distributed under probability $\mathbb{P}$, then the conformal set $\Gamma^{(\alpha,\epsilon)}(x_{n+1})$ satisfies the coverage guarantee $\mathbb{P}^{n+1}(y_{n+1} \in \Gamma^{(\alpha,\epsilon)}(x_{n+1})) \geq 1 - \alpha$.*

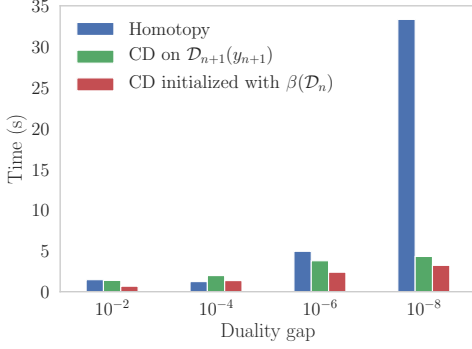

| | Coverage | Length | Time (s) |
|---|---|---|---|
| Oracle | 0.9 | 1.685 | 0.59 |
| Split | 0.9 | 3.111 | 0.26 |
| 1e-2 | 0.9 | 1.767 | 2.17 |
| 1e-4 | 0.9 | 1.727 | 8.02 |
| 1e-6 | 0.9 | 1.724 | 45.94 |
| 1e-8 | 0.9 | 1.722 | 312.56 |

Table 1: Computing a conformal set for a Lasso regression problem on a climate data set `NCEP/NCAR Reanalysis` [11] with $n = 814$ observations and $p = 73570$ features. On the left, we compare the time needed to compute the full approximation path with our homotopy strategy, single coordinate descent (CD) on the full data $\mathcal{D}_{n+1}(y_{n+1})$, and an update of the solution after initialization with an approximate solution using $\mathcal{D}_n$. On the right, we display the coverage, length and time of different methods averaged over 100 randomly held-out validation data sets.

The conformal prediction set $\Gamma^{(\alpha,\epsilon)}(x_{n+1})$ (with an approximate solution) preserves the $1 - \alpha$ coverage guarantee and converges to $\Gamma^{(\alpha,0)}(x_{n+1}) = \hat{\Gamma}^{(\alpha)}(x_{n+1})$ (with an exact solution) when the optimization error decreases to zero. It is also easier to compute in the sense that only a finite number of candidates $z$ need to be evaluated. Indeed, as soon as an approximate solution $\beta(z)$ is allowed, we have shown in Section 3 that a solution update is not necessary for neighboring observation candidates.

We consider the parameterization in Equation (9). It holds that

$$\Gamma^{(\alpha,\epsilon)} = \{z \in \mathbb{R} : \pi(z, \epsilon) > \alpha\} = \{z_t : t \in \mathbb{R}, \pi(z_t, \epsilon) > \alpha\} \ .$$

Using Algorithm 1, we can build a set $\{z_{t_1}, \cdots, z_{t_{T_\epsilon}}\}$ that covers $[y_{\min}, y_{\max}]$ with $\epsilon$-solutions *i.e.* :

$$\forall z \in [y_{\min}, y_{\max}], \exists k \in [T_\epsilon] \text{ such that } \text{Gap}_z(\beta(z_{t_k}), \theta(z_{t_k})) \leq \epsilon \ .$$

Using the classical conformity measure $\hat{R}_i(z) = |y_i - x_i^\top \hat{\beta}(z)|$ and computing a piecewise constant approximation of the solution path $t \mapsto \hat{\beta}(z_t)$ with the set $\{\beta(z_{t_k}) : k \in [T_\epsilon]\}$, we have

$$\Gamma^{(\alpha,\epsilon)} \cap [y_{\min}, y_{\max}] = \bigcup_{k \in [T_\epsilon]} [z_{t_k}, z_{t_{k+1}}] \cap [x_{n+1}^\top \beta(z_{t_k}) \pm Q_{1-\alpha}(z_{t_k})] \ .$$

where $Q_{1-\alpha}(z)$ is the $(1 - \alpha)$-quantile of the sequence of approximate residuals $(R_i(z))_{i \in [n+1]}$.

Details and extensions to the more general cases of conformity measures are discussed in the appendix.

## 4.2 Wrapping the Exact Conformal Set

Previously, we showed that a full conformal set can be efficiently computed with an approximate solution, and it converges to the conformal set with an exact solution when the optimization error decreases to zero. When the loss function is smooth and, under a gradient-based conformity measure (introduced below), we provide a stronger guarantee that the exact conformal set can be included in a conformal set, using only approximate solutions. For this, we show how the conformity measure can be bounded *w.r.t.* to the optimization error, when the input observation $z$ changes.

**Gradient based Conformity Measures.** The separability of the loss function implies that the coordinate-wise absolute value of the gradient of the loss function preserves the excheangeability of the data, and then the coverage guarantee. Whence it can be safely used as a conformity measure *i.e.*

$$\hat{R}_:(z) = |\nabla \ell(Y_z, X\hat{\beta}(z))|, \qquad\qquad R_:(z) = |\nabla \ell(Y_z, X\beta(z))| \ . \qquad (13)$$

Using Equation (13), we show how the function $\hat{\pi}$ can be approximated from above and below, thanks to a fine bound on the dual optimal solution [15], which is related to the gradient of the loss function.

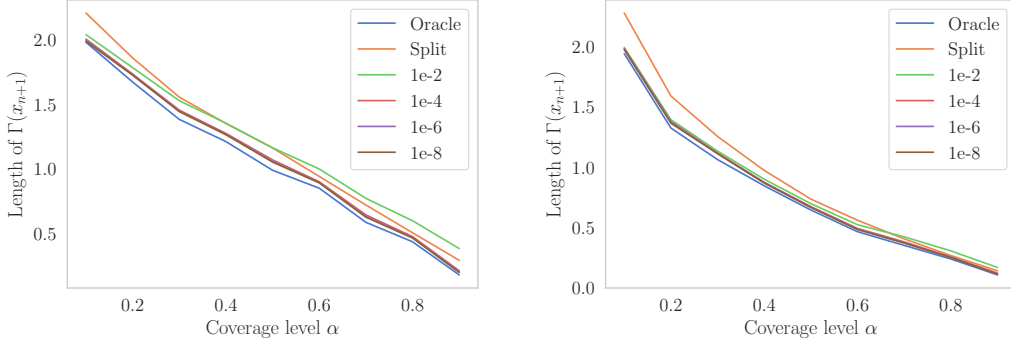

(a) Linear regression with $\ell_1$ regularization on Diabetes dataset ($n = 442, p = 10$).

(b) `Logcosh` regression with $\ell_2^2$ regularization on Boston dataset ($n = 506, p = 13$).

Figure 2: Length of the conformal prediction sets at different coverage level $\alpha \in \{0.1, 0.2, \cdots, 0.9\}$. For all $\alpha$, we display the average over 100 repetitions of randomly held-out validation data sets.

**Lemma 2.** *If the loss function $\ell(z, \cdot)$ is $\nu$-smooth, for any real value $z$, we have*

$$\|\theta(z) - \hat{\theta}(z)\|^2 \le \frac{2\nu}{\lambda^2} \operatorname{Gap}_z(\beta(z), \theta(z)), \quad \forall (\beta(z), \theta(z)) \in \operatorname{dom}P_z \times \operatorname{dom}D_z .$$

Using Equation (13) and further assuming that the dual vector $\theta(z)$ constructed in Equation (10) coincides [1] with $-\nabla\ell(Y_z, X\beta(z))/\lambda$ in $\operatorname{dom}D_z$, we have $\hat{R}_:(z) = \|\lambda\hat{\theta}(z)\|$ and $R_:(z) = \|\lambda\theta(z)\|$. Thus, combining the triangle inequality and Lemma 2 we have

$$\forall i \in [n+1], \ (R_i(z) - \hat{R}_i(z))^2 \le \|R_:(z) - \hat{R}_:(z)\|^2 = \lambda^2\|\theta(z) - \hat{\theta}(z)\|^2 \le 2\nu\epsilon ,$$

where the last inequality holds as soon as we can maintain $\operatorname{Gap}_z(\beta(z), \theta(z))$ to be smaller than $\epsilon$, for any $z$ in $\mathbb{R}$. Whence, $\hat{R}_i(z)$ belongs to $[R_i(z) \pm \sqrt{2\nu\epsilon}]$ for any $i$ in $[n+1]$. Noting that

$$\hat{\pi}(z) = 1 - \frac{1}{n+1}\operatorname{Rank}(\hat{R}_{n+1}(z)) = \frac{1}{n+1}\sum_{i=1}^{n+1}\mathbb{1}_{\hat{R}_i(z) \ge \hat{R}_{n+1}(z)} ,$$

the function $\hat{\pi}$ can be easily approximated from above and below by the functions $\underline{\pi}(z, \epsilon)$ and $\overline{\pi}(z, \epsilon)$, which do not depend on the exact solution and are defined as:

$$\underline{\pi}(z, \epsilon) = \frac{1}{n+1}\sum_{i=1}^{n+1}\mathbb{1}_{R_i(z) \ge R_{n+1}(z) + 2\sqrt{2\nu\epsilon}}, \quad \overline{\pi}(z, \epsilon) = \frac{1}{n+1}\sum_{i=1}^{n+1}\mathbb{1}_{R_i(z) \ge R_{n+1}(z) - 2\sqrt{2\nu\epsilon}} .$$

**Proposition 3.** *We assume that the loss function is $\nu$-smooth and that we use a gradient based conformity measure (13). Then, we have $\underline{\pi}(z, \epsilon) \le \hat{\pi}(z) \le \overline{\pi}(z, \epsilon)$ and the approximated lower and upper bounds of the exact conformal set are $\underline{\Gamma}^{(\alpha, \epsilon)} \subset \hat{\Gamma}^{(\alpha)} \subset \overline{\Gamma}^{(\alpha, \epsilon)}$ where*

$$\underline{\Gamma}^{(\alpha, \epsilon)} = \{z \in \mathbb{R} : \underline{\pi}(z, \epsilon) > \alpha\}, \quad\quad \overline{\Gamma}^{(\alpha, \epsilon)} = \{z \in \mathbb{R} : \overline{\pi}(z, \epsilon) > \alpha\} .$$

In the baseline case of quadratic loss, such sets can be easily computed as

$$\overline{\Gamma}^{(\alpha, \epsilon)} \cap [y_{\min}, y_{\max}] = \bigcup_{k \in [T_\epsilon]} [z_{t_k}, z_{t_{k+1}}] \cap [x_{n+1}^\top\beta(z_{t_k}) \pm Q_{1-\alpha}^-(t_k)] ,$$

$$\underline{\Gamma}^{(\alpha, \epsilon)} \cap [y_{\min}, y_{\max}] = \bigcup_{k \in [T_\epsilon]} [z_{t_k}, z_{t_{k+1}}] \cap [x_{n+1}^\top\beta(z_{t_k}) \pm Q_{1-\alpha}^+(t_k)] ,$$

where we have denoted $Q_{1-\alpha}^-(t_k)$ (resp. $Q_{1-\alpha}^+(t_k)$) as the $(1-\alpha)$-quantile of the sequence of shifted approximate residuals $(R_i(z_{t_k}) - 2\sqrt{2\nu\epsilon})_{i \in [n+1]}$ (resp. $(R_i(z_{t_k}) + 2\sqrt{2\nu\epsilon})_{i \in [n+1]}$) corresponding to the approximate solution $\beta(z_{t_k})$ for $k$ in $[T_\epsilon]$.

| | Oracle | Split | 1e-2 | 1e-4 | 1e-6 | 1e-8 |
|---|---|---|---|---|---|---|
| Smooth Chebychev Approx. | | | | | | |
| Coverage | 0.92 | 0.95 | 0.92 | 0.92 | 0.92 | 0.92 |
| Length | 1.940 | 2.271 | 1.998 | 1.990 | 1.987 | 1.981 |
| Time (s) | 0.019 | 0.016 | 0.073 | 0.409 | 3.742 | 36.977 |
| | | | | | | |
| Linex regression | | | | | | |
| Coverage | 0.91 | 0.93 | 0.91 | 0.91 | 0.91 | 0.91 |
| Length | 2.189 | 2.447 | 2.231 | 2.209 | 2.205 | 2.199 |
| Time (s) | 0.013 | 0.012 | 0.050 | 0.234 | 2.054 | 20.712 |

Table 2: Computing a conformal set for a `logcosh` (resp. `linex`) regression problem regularized with a Ridge penalty on the Boston (resp. Diabetes) dataset with $n = 506$ observations and $p = 13$ features (resp. $n = 442$ and $p = 10$). We display the coverage, length and time of the different methods, averaged over 100 randomly held-out validation data sets.

# 5 Numerical Experiments

We illustrate the approximation of a full conformal prediction set for both linear and non-linear regression problems, using synthetic and real datasets that are publicly available in `sklearn`. All experiments were conducted with a coverage level of $0.9$ ($\alpha = 0.1$) and a regularization parameter selected by cross-validation on a randomly separated training set (for real data, we used $33\%$ of the data).

In the case of Ridge regression, *exact* and *full* conformal prediction sets can be computed without any assumptions [18]. We show in Figure 1, the conformal sets *w.r.t.* different regularization parameters $\lambda$, and our proposed method based on an approximated solution for different optimization errors. The results indicate that high precision is not necessary to obtain a conformal set close to the exact one.

For other problem formulations, we define an `Oracle` as the set $[x_{n+1}^\top \hat{\beta}(y_{n+1}) \pm \hat{Q}_{1-\alpha}(y_{n+1})]$ obtained from the estimator trained with machine precision on the oracle data $\mathcal{D}_{n+1}(y_{n+1})$ (the target variable $y_{n+1}$ is not available in practice). For comparison, we display the average over 100 repetitions of randomly held-out validation data sets, the empirical coverage guarantee, the length, and time needed to compute the conformal set with splitting and with our approach.

We illustrated in Table 1 the computational cost of our proposed homotopy for Lasso regression, using vanilla coordinate descent (CD) optimization solvers in `sklearn` [19]. For a large range of duality gap accuracies $\epsilon$, the computational time of our method is roughly the same as a single run of CD on the full data set. However, when $\epsilon$ becomes very small ($\approx 10^{-8}$), we lose computational time efficiency due to large complexity $T_\epsilon$. This is visible in regression problems with non-quadratic loss functions Table 2.

The computational times depend only on the data fitting part and the computation of the conformity score functions. Thus, the computational efficiency is independent of the coverage level $\alpha$. We show in Figure 2, the variations of the length of the conformal prediction set for different coverage level.

Overall, the results indicate that the homotopy method provides valid and near-perfect coverage, regardless of the optimization error $\epsilon$. The lengths of the confidence sets generated by homotopy methods gradually increase as $\epsilon$ increases, but all of the sets are consistently smaller than those of splitting approaches. Our experiments showed that high accuracy has only limited benefits.

**Acknowledgments**

We would like to thank the reviewers for their valuable feedbacks and detailed comments which contributed to improve the quality of this paper. This work was partially supported by MEXT KAKENHI (17H00758, 16H06538), JST CREST (JPMJCR1502), RIKEN Center for Advanced Intelligence Project, and JST support program for starting up innovation-hub on materials research by information integration initiative.

## Footnotes

[1] This holds whenever $\Omega$ is strongly convex or its domain is bounded. Also, one can guarantee this condition when $\beta(z)$ is build using any converging iterative algorithm, with sufficient iterations, for solving Equation (3).

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
