[Supplementary Material]

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

# 6 Appendix

**More examples of Loss Function.** Popular instances of loss functions can be found in the literature. For instance, in `power norm regression`, $\ell(a,b) = |a - b|^q$. When $q = 2$, it corresponds to classical linear regression and the cases where $q \in [1, 2)$ are common in robust statistics. In particular $q = 1$ is known as least absolute deviation. One can also have the `log-cosh` loss $\ell(a,b) = \gamma \log(\cosh(a-b)/\gamma)$ as a differentiable alternative for the $\ell_\infty$ norm (chebychev approximation). One also have the `Linex` loss function [10, 5] which provide an asymmetric loss $\ell(a,b) = \exp(\gamma(a - b)) - \gamma(a - b) - 1$, for $\gamma \neq 0$. Least square fitting with non linear transformation where the relation between the observations and the features are described as $y_i \approx \phi(x_i, \beta)$ where $\phi$ is derived from physical or biological prior knowledge on the data. For instances, we have the exponential model $\phi(x_i, \beta) = a \exp(bx_i)$. Any convex regularization functions $\Omega$ can be considered. Popular examples are sparsity inducing norm [2], Ridge [12], elastic net [28], total variation, $\ell_\infty$, sorted $\ell_1$ norm etc.

## 6.1 Homotopy with Different Regularity

We recall that from Lemma 1, we have

$$\Delta G(x_{n+1}, z_t, z_0) = [\ell(z, x_{n+1}^\top \beta) - \ell(z_0, x_{n+1}^\top \beta)] + [\ell^*(z_t, -\lambda\theta_{n+1}) - \ell^*(z_0, -\lambda\theta_{n+1})] \ . \quad (14)$$

We also recall the assumptions on the loss function.

**Assumption A1.** The functions $\ell$ and $\Omega$ are bounded from below. Thus, without loss of generality, we can also assume for any real value $z_0$ that $\ell^*(z_0, 0) = -\inf_z \ell(z_0, z) = 0$ otherwise one can always replace $\ell(z_0, \cdot)$ by $\ell(z_0, \cdot) - \inf_z \ell(z_0, z)$.

**Assumption A2.** For any real values $z$ and $z_0$, we have $\ell(z_0, z), \ell(z, z_0) \geq 0$ and $\ell(z_0, z_0) = 0$.

This assumptions helps to simplify the first order expansion of $\ell$ at $z_0$ since $z_0 = \arg\min_z \ell(z, z_0)$ which is equivalent to $\partial_1 \ell(z_0, z_0) = 0$. Similarly, we also have $\partial_2 \ell(z_0, z_0) = 0$

Now we apply the formula Equation (14) to $z_0 = x_{n+1}^\top \beta$ and $z_t = z_0 + t$. Furthermore, using the dual vector in Equation (10), we have by construction $\theta_{n+1} \propto \partial_2 \ell(z_0, x_{n+1}^\top \beta) = \partial_2 \ell(z_0, z_0) = 0$. Then the variation of the gap between $z_t$ and $z_0$ simplifies to

$$\Delta G(x_{n+1}, z_t, z_0) = \ell(z_t, z_0) \ . \quad (15)$$

**Smooth Loss.** To simplify the notation, given a real value $b$, we denote $\ell_{[b]}(a) = \ell(a, b)$ which is assumed to be a $\nu$-smooth function *i.e.*

$$\ell_{[b]}(a) \leq \ell_{[b]}(a_0) + \langle \ell'_{[b]}(a_0), a - a_0 \rangle + \frac{\nu}{2}(a - a_0)^2, \quad \forall a, a_0 \ . \quad (16)$$

By assumption, $\ell_{[b]}(b) = 0$ and $\ell_{[b]}(a) \geq 0$. Thus we have $b = \arg\min_a \ell_{[b]}(a)$ which implies $\ell'_{[b]}(b) = 0$. Then $\ell(a,b) = \ell_{[b]}(a) \leq \frac{\nu}{2}(a - b)^2$; applied to $a = z_t$ and $b = z_0$, it reads:

$$\Delta G(x_{n+1}, z_t, z_0) \leq \frac{\nu}{2}(z_{n+1}(t) - z_0)^2 = \frac{\nu}{2}t^2 \ . \quad (17)$$

**Lipschitz Loss.** We suppose that the loss function is $\nu$-Lipschitz *i.e.*

$$|\ell_{[b]}(a) - \ell_{[b]}(a_0)| \leq \nu|a - a_0| \ . \quad (18)$$

Applying Equation (18) to $a = z_t$ and $b = a_0 = z_0$ reads:

$$\Delta G(x_{n+1}, z_t, z_0) \leq \nu|z_t - z_0| = \nu|t| \ .$$

Whence the variation of the gap $\Delta G(x_{n+1}, z_t, z_0)$ are smaller than $\epsilon$ as soon as $t \in [-\epsilon/\nu, \epsilon/\nu]$. In that case, the complexity of the homotopy for covering the interval $[y_{\min}, y_{\max}]$ is

$$T_\epsilon \leq \left\lceil \frac{y_{\max} - y_{\min}}{\epsilon/\nu} \right\rceil \in O\left(\frac{1}{\epsilon}\right) \ .$$

**$\mathcal{V}$-smooth Loss.**  We suppose that the loss function is uniformly smooth *i.e.*

$$\ell_{[b]}(a) \le \ell_{[b]}(a_0) + \langle \ell'_{[b]}(a_0), a - a_0 \rangle + \mathcal{V}_{a_0}(a - a_0), \quad \forall a, a_0 \ , \tag{19}$$

where $\mathcal{V}_{a_0}$ is a non negative functions vanishing at zero.

Applying Equation (19) to $a = z_t$ and $b = a_0 = z_0$ reads:

$$\Delta G(x_{n+1}, z_t, z_0) \le \mathcal{V}_{z_0}(z_t - z_0) = \mathcal{V}_{z_0}(t) \ . \tag{20}$$

The $\mathcal{V}$-smooth regularity contains two important known cases of local and global smoothness with different order:

- **Uniformly Smooth Loss** [1]. In this case, $\mathcal{V}_{a_0}(a - a_0) = \mathcal{V}(\|a - a_0\|)$ does not depends on $a_0$ and where $\mathcal{V}$ is any non increasing function from $[0, +\infty)$ to $[0, +\infty]$ *e.g.* $\mathcal{V}(t) = \frac{\mu}{d} t^d$. When $d = 2$, we recover the classical smoothness in (16).

  Thus, the variation of the gap $\Delta G(x_{n+1}, z_t, z_0)$ are smaller than $\epsilon$ as soon as $t \in [-\mathcal{V}^{-1}(\epsilon), \mathcal{V}^{-1}(\epsilon)]$. This leads to a generalized complexity of the homotopy for covering $[y_{\min}, y_{\max}]$ in $T_\epsilon$ steps where

$$T_\epsilon \le \left\lceil \frac{y_{\max} - y_{\min}}{\mathcal{V}^{-1}(\epsilon)} \right\rceil \in O\left( \frac{1}{\mathcal{V}^{-1}(\epsilon)} \right) \ .$$

- **Generalized Self-Concordant Loss** [26]. A $\mathcal{C}^3$ convex function $f$ is $(M_f, \nu)$-generalized self-concordant of order $\nu \ge 2$ and $M_f \ge 0$ if $\forall x \in \mathrm{dom} f$ and $\forall u, v \in \mathbb{R}^n$:

$$\left| \langle \nabla^3 f(x)[v]u, u \rangle \right| \le M_f \|u\|_x^2 \|v\|_x^{\nu-2} \|v\|_2^{3-\nu} \ .$$

In this case, [26, Proposition 10] have shown that one could write:

$$\mathcal{V}_{\ell_{[b]}, a_0}(a - a_0) = w_\nu(d_\nu(a_0, a)) \|a - a_0\|_{a_0}^2 \ ,$$

where the last equality holds if $d_\nu(a_0, a) < 1$ for the case $\nu > 2$. Closed-form expressions of $w_\nu(\cdot)$ and $d_\nu(\cdot)$ are given as follow:

$$d_\nu(a_0, a) := \begin{cases} M_{\ell_{[b]}} \|a - a_0\|_2 & \text{if } \nu = 2, \\ \left(\frac{\nu}{2} - 1\right) M_{\ell_{[b]}} \|a - a_0\|_2^{3-\nu} \|a - a_0\|_{a_0}^{\nu-2} & \text{if } \nu > 2, \end{cases} \tag{21}$$

and

$$w_\nu(\tau) := \begin{cases} \frac{e^\tau - \tau - 1}{\tau^2} & \text{if } \nu = 2, \\ \frac{-\tau - \log(1 - \tau)}{\tau^2} & \text{if } \nu = 3, \\ \frac{(1-\tau)\log(1-\tau) + \tau}{\tau^2} & \text{if } \nu = 4, \\ \left(\frac{\nu-2}{4-\nu}\right) \frac{1}{\tau} \left[ \frac{\nu-2}{2(3-\nu)\tau} \left( (1-\tau)^{\frac{2(3-\nu)}{2-\nu}} - 1 \right) - 1 \right] & \text{otherwise.} \end{cases} \tag{22}$$

Power loss function $\ell_{[b]}(a, b) = (a - b)^q$ for $q \in (1, 2)$, popular in robust regression, is covered with $M_{\ell_{[b]}} = \frac{2-q}{(2-q)\sqrt{q(q-1)}}, \nu = \frac{2(3-q)}{2-q} \in (4, +\infty)$.

We refer to [26] for more details and examples.

Note that when local smoothness is used, the step sizes depend on the current candidate $z_{t_k}$ along the path: the generated grid is adaptive and the step sizes can be computed numerically.

## 6.2  Proofs

**Lemma 3** (c.f. Lemma 1). *For any $(\beta, \theta) \in \mathrm{dom} P_z \times \mathrm{dom} D_z$ for $z \in \{z_0, y\}$, we have*

$$\Delta G(x_{n+1}, z, z_0) = [\ell(z, x_{n+1}^\top \beta) - \ell(z_0, x_{n+1}^\top \beta)] + [\ell^*(z, -\lambda\theta_{n+1}) - \ell^*(z_0, -\lambda\theta_{n+1})] \ . \tag{23}$$

*Proof.* By definition,

$$\Delta G(x_{n+1}, z, z_0) = \mathrm{Gap}_z(\beta, \theta) - \mathrm{Gap}_{z_0}(\beta, \theta) = [P_z(\beta) - D_z(\theta)] - [P_{z_0}(\beta) - D_{z_0}(\theta)]$$
$$= [P_z(\beta) - P_{z_0}(\beta)] - [D_z(\theta) - D_{z_0}(\theta)] \ .$$

The conclusion follows from the fact that the first term is

$$P_z(\beta) - P_{z_0}(\beta) = \ell(z, x_{n+1}^\top \beta) - \ell(z_0, x_{n+1}^\top \beta) \ ,$$

and the second term is

$$D_z(\theta) - D_{z_0}(\theta) = \ell^*(z_0, -\lambda\theta_{n+1}) - \ell^*(z, -\lambda\theta_{n+1}) \ .$$

$\square$

For the initialization, we start with a couple of vector $(\beta, \theta) \in \mathrm{dom}P \times \mathrm{dom}D \subset \mathbb{R}^p \times \mathbb{R}^n$ that we need to extent to $(\beta, \theta^+) \in \mathrm{dom}P_z \times \mathrm{dom}D_z \subset \mathbb{R}^p \times \mathbb{R}^{n+1}$. For better clarity, we restate the previous lemma to this specific case.

**Lemma 4.** *Let $(\beta, \theta)$ be any primal/dual vector in $\mathrm{dom}P \times \mathrm{dom}D$ and $\theta^+ = (\theta, 0)$ in $\mathbb{R}^{n+1}$. For any real value $z$, the variation of the duality gap is equal to the loss between $z$ and $x_{n+1}^\top \beta$ i.e.*

$$\Delta G(x_{n+1}, z) := \mathrm{Gap}_z(\beta, \theta^+) - G(\beta, \theta) = \ell(z, x_{n+1}^\top \beta) \ .$$

*Proof.* Let $\delta \in \mathbb{R}$ such that $\theta_\delta^+ = (\theta, \delta)^\top \in \mathbb{R}^{n+1} \in \mathrm{dom}D_z$. We have

$$
\begin{aligned}
\Delta G(\delta) &:= \mathrm{Gap}_z(\beta, \theta_\delta^+) - G(\beta, \theta) \\
&= [P_z(\beta) - P(\beta)] - [D_z(\theta_\delta^+) - D(\theta)] \\
&= \ell(z, x_{n+1}^\top \beta) + \ell^*(z, -\lambda\delta) + \lambda [\Omega^*(X_{[n]}^\top \theta + \delta x_{n+1}^\top) - \Omega^*(X_{[n]}^\top \theta)] \ .
\end{aligned}
$$

We choose $\delta = 0$; which is admissible because $0 \in \mathrm{dom}\ell^*(z, \cdot)$ if and only if $\ell(z, \cdot)$ is bounded from below as assumed. The result follows the observation that $\Delta G(0) = \Delta G(x_{n+1}, z)$. $\square$

**Proposition 4** (c.f. Proposition 2). *Given a significance level $\alpha \in (0, 1)$ and an optimization tolerance $\epsilon > 0$, if the observations $(x_i, y_i)_{i \in [n+1]}$ are exchangeable and identically distributed under probability $\mathbb{P}$, then the conformal set $\Gamma^{(\alpha, \epsilon)}(x_{n+1})$ satisfies the coverage guarantee*

$$\mathbb{P}^{n+1}(y_{n+1} \in \Gamma^{(\alpha, \epsilon)}(x_{n+1})) \geq 1 - \alpha \ .$$

*Proof.* The separability of the loss function in $P_z$ implies that

$$\beta((x_i, y_i)_{i \in [n+1]}) = \beta((x_{\sigma(i)}, y_{\sigma(i)})_{i \in [n+1]}) \ ,$$

for any permutation $\sigma$ of the index set $\{1, \cdots, n+1\}$. Whence the sequence of conformity measure $(R_i(y_{n+1}))_{i \in [n+1]}$ is invariant *w.r.t.* permutation of the data.

The exchangeability of the sequence $\{(x_i, y_i)_{i \in [n+1]}\}$ implies that of $(R_i(y_{n+1}))_{i \in [n+1]}$.

The rests of the proof are based on the fact that the rank of one variable among an *exchangeable and identically distributed* sequence is (sub)-uniformly distributed [4].

**Lemma 5.** *Let $U_1, \ldots, U_{n+1}$ be exchangeable and identically distributed sequence of real valued random variables. Then for any $\alpha \in (0, 1)$, we have $\mathbb{P}^{n+1}(\mathrm{Rank}(U_{n+1}) \leq (n+1)(1-\alpha)) \geq 1 - \alpha$.*

Using Lemma 5, we deduce that the rank of $R_{n+1}(y_{n+1})$ among $(R_i(y_{n+1}))_{i \in [n+1]}$ is sub-uniformly distributed on the discrete set $\{1, \cdots, n+1\}$. Recalling the definition of *typicalness*

$$\forall z \in \mathbb{R}, \quad \pi(z, \epsilon) = 1 - \frac{1}{n+1}\mathrm{Rank}(R_{n+1}(z)) \ ,$$

We have

$$\mathbb{P}^{n+1}(\pi(y_{n+1}, \epsilon) > \alpha) = \mathbb{P}^{n+1}(\mathrm{Rank}(R_{n+1}(y_{n+1}) < (n+1)(1-\alpha)) \geq 1 - \alpha \ .$$

The proof for conformal set with exact solution corresponds to $\epsilon = 0$. $\square$

**Lemma 6** (c.f. Lemma 2). *Assuming that $\ell(y_i, \cdot)$ is $\nu$-smooth, we have*

$$\|\theta(z) - \hat\theta(z)\|^2 \leq \frac{2\nu}{\lambda^2} \mathrm{Gap}_z(\beta(z), \theta(z)) \ . \tag{24}$$

Note that such a bound on the dual optimal solution, leveraging duality gap, was used in optimization [17, 25] to bound the Lagrange multipliers for identifying sparse components in lasso type problems.

*Proof.* Remember that $\forall i \in [n], \ell(y_i, \cdot)$ is $\nu$-smooth. As a consequence, $\forall i \in [n], \ell^*(y_i, \cdot)$ is $1/\nu$-strongly convex [11, Theorem 4.2.2, p. 83] and so the dual function $D_\lambda$ is $\lambda^2/\nu$-strongly concave:

$$\forall (\theta_1, \theta_2) \quad D_z(\theta_2) \le D_z(\theta_1) + \langle \nabla D_z(\theta_1), \theta_2 - \theta_1 \rangle - \frac{\lambda^2}{2\nu} \|\theta_1 - \theta_2\|^2 \ .$$

Specifying the previous inequality for $\theta_1 = \hat{\theta}(z), \theta_2 = \theta(z)$, one has

$$D_z(\theta) \le D_z(\hat{\theta}(z)) + \langle \nabla D_z(\hat{\theta}(z)), \theta(z) - \hat{\theta}(z) \rangle - \frac{\lambda^2}{2\nu} \|\hat{\theta}(z) - \theta(z)\|^2 \ .$$

By definition, $\hat{\theta}(z)$ maximizes $D_z$, so, $\langle \nabla D_z(\hat{\theta}(z)), \theta(z) - \hat{\theta}(z) \rangle \le 0$. This implies

$$D_z(\theta(z)) \le D_z(\hat{\theta}(z)) - \frac{\lambda^2}{2\nu} \|\hat{\theta}(z) - \theta(z)\|^2.$$

By weak duality, we have $D_z(\hat{\theta}(z)) \le P_z(\beta(z))$, hence

$$D_z(\theta(z)) \le P_z(\beta(z)) - \frac{\lambda^2}{2\nu} \|\hat{\theta}(z) - \theta(z)\|^2$$

and the conclusion follows. $\qquad\square$

**Proposition 5** (c.f. Proposition 3)**.** *We assume that the loss function is $\nu$-smooth and that we use a gradient based conformity measure* (13)*. Then, we have $\underline{\pi}(z, \epsilon) \le \hat{\pi}(z) \le \overline{\pi}(z, \epsilon)$ and the approximated lower and upper bounds of the exact conformal set are $\underline{\Gamma}^{(\alpha, \epsilon)} \subset \hat{\Gamma}^{(\alpha)} \subset \overline{\Gamma}^{(\alpha, \epsilon)}$ where*

$$\underline{\Gamma}^{(\alpha, \epsilon)} = \{z \in \mathbb{R} : \underline{\pi}(z, \epsilon) > \alpha\}, \qquad \overline{\Gamma}^{(\alpha, \epsilon)} = \{z \in \mathbb{R} : \overline{\pi}(z, \epsilon) > \alpha\} \ .$$

*Proof.* We recall that for any $i$ in $[n+1]$, we have $\hat{R}_i(z)$ belongs to $[R_i(z) \pm \sqrt{2\nu\epsilon}]$. Then

$$\hat{R}_i(z) \ge \hat{R}_{n+1}(z) \implies R_i(z) + \sqrt{2\nu\epsilon} \ge \hat{R}_i(z) \ge \hat{R}_{n+1}(z) \ge R_{n+1}(z) - \sqrt{2\nu\epsilon}$$
$$\implies R_i(z) \ge R_{n+1}(z) - 2\sqrt{2\nu\epsilon} \ .$$

Whence $\hat{\pi}(z) \le \overline{\pi}(z, \epsilon)$. The inequality $\underline{\pi}(z, \epsilon) \le \hat{\pi}(z)$ follows from the fact that $R_i(z) - \sqrt{2\nu\epsilon}$ (resp. $R_{n+1}(z) + \sqrt{2\nu\epsilon}$) is a lower bound of $\hat{R}_i(z)$ (resp. upper bound of $\hat{R}_{n+1}(z)$). $\qquad\square$

### 6.3 Details on Practical Computations

For simplicity, let us first restrict to the case of quadratic loss where the conformity measure is defined such as $\hat{R}_i(z) = |y_i - x_i^\top \hat{\beta}(z)|$.

Note that $\pi(z, \epsilon) > \alpha$ if and only if $R_{n+1}(z) \le Q_{1-\alpha}(z)$ where $Q_{1-\alpha}(z)$ is the $(1 - \alpha)$-quantile of the sequence of approximate residual $(R_i(z))_{i \in [n+1]}$. Then the approximate conformal set can be conveniently written as

$$\Gamma^{(\alpha, \epsilon)} = \{z \in \mathbb{R} : R_{n+1}(z) \le Q_{1-\alpha}(z)\} = \bigcup_{z \in \mathbb{R}} [x_{n+1}^\top \beta(z) \pm Q_{1-\alpha}(z)] \ .$$

Let $\{\beta(z_{t_k}) : k \in [T_\epsilon]\}$ be the set of solutions outputted by the approximation homotopy method, the functions $t \mapsto \beta(z_t)$ $\epsilon$-solution of optimization problem (3) using the data $\mathcal{D}_{n+1}(z_t)$ and $t \mapsto x^\top \beta(z_t)$, are piecewise constant on the intervals $(t_k, t_{k+1})$. Also, the map $t \mapsto R_{n+1}(z_t)$ (resp. $t \mapsto R_i(z_t)$ for $i$ in $[n]$) is piecewise linear (resp. piecewise constant) on $[t_k, t_{k+1}]$. Thus, we have

$$\Gamma^{(\alpha, \epsilon)} \cap [y_{\min}, y_{\max}] = \{z_t : t \in \mathbb{R}, R_{n+1}(z_t) \le Q_{1-\alpha}(z_t\} \cap [y_{\min}, y_{\max}]$$
$$= \bigcup_{k \in [T_\epsilon]} \{z_t : t \in [t_k, t_{k+1}], R_{n+1}(z_t) \le Q_{1-\alpha}(z_t)\}$$
$$= \bigcup_{k \in [T_\epsilon]} [z_{t_k}, z_{t_{k+1}}] \cap [x_{n+1}^\top \beta(z_{t_k}) \pm Q_{1-\alpha}(z_{t_k})] \ .$$

where $Q_{1-\alpha}(z)$ is the $(1 - \alpha)$-quantile of the sequence of approximate residual $(R_i(z))_{i \in [n+1]}$.

**Extensions to Others Nonconformity Measure.** We consider a generic conformity measure in Equation (4). We basically follow the same step than the derivation of conformal set for ridge in [27].

For any $i$ in $[n+1]$, we denote the intersection points of the functions $R_i(z_t)$ and $R_{n+1}(z_t)$ restricted on the interval $[t_k, t_{k+1}]$: $t_{k,i}^-$ and $t_{k,i}^+$.

We however assume that there is only two intersection points. The computations for more finitely points are the same. For instance, using the absolute value as conformity measure, we have

$$t_{k,i}^- = (\mu_{t_k}(x_{n+1}) - R_i(z_{t_k}) - z_0), \qquad t_{k,i}^+ = (\mu_{t_k}(x_{n+1}) + R_i(z_{t_k}) - z_0) \ ,$$

where $\mu_{t_k}(x_{n+1}) := x_{n+1}^\top \beta(z_{t_k})$. Now, let us define

$$S_i = \{t \in [t_{\min}, t_{\max}] : R_i(z_t) \geq R_{n+1}(z_t)\} = \bigcup_{k \in [T_\epsilon]} S_i \cap [t_k, t_{k+1}] = \bigcup_{k \in [T_\epsilon]} [t_{k,i}^-, t_{k,i}^+] \ .$$

For any $k$ in $[T_\epsilon]$, we denote the set of solutions $t_{k,1}^-, t_{k,1}^+, \cdots, t_{k,n+1}^-, t_{k,n+1}^+$ in increasing order as $t_k = t_{k,0} < t_{k,1} < \cdots < t_{k,l_k} = t_{k+1}$. Whence for any $t \in [t_k, t_{k+1}]$, it exists a unique index $j = \mathcal{J}(t)$ such that $t \in (t_{k,j}, t_{k,j+1})$ or $t \in \{t_{k,j}, t_{k,j+1}\}$ and for any $t \in [t_k, t_{k+1}]$, we have

$$(n+1)\pi(z_t) = \sum_{i=1}^{n+1} \mathbb{1}_{t \in S_i \cap [t_k, t_{k+1}]} = N_k(\mathcal{J}(t)) + M_k(\mathcal{J}(t))$$

where the functions

$$N_k(j) = \sum_{i=1}^{n+1} \mathbb{1}_{(t_{k,j}, t_{k,j+1}) \subset [t_{k,i}^-, t_{k,i}^+]} \text{ and } M_k(j) = \sum_{i=1}^{n+1} \mathbb{1}_{t_{k,j} \in [t_{k,i}^-, t_{k,i}^+]}$$

Note that $\mathcal{J}^{-1}([t_k, t_{k+1}]) = \{0, 1, \cdots, l_k\}$ and $\mathcal{J}^{-1}(j) = [t_{k,j}, t_{j+1}]$. Finally, we have

$$\Gamma^{(\alpha, \epsilon)} \cap [y_{\min}, y_{\max}] = \bigcup_{k \in [T_\epsilon]} \Gamma^{(\alpha, \epsilon)} \cap [t_k, t_{k+1}] \tag{25}$$

$$= \bigcup_{k \in [T_\epsilon]} \bigcup_{\substack{j \in [0:l_k] \\ N_k(j) > (n+1)\alpha}} (t_{k,j}, t_{k,j+1}) \ \cup \bigcup_{\substack{j \in [0:l_k] \\ M_k(j) > (n+1)\alpha}} \{t_j\} \ . \tag{26}$$

## 6.4 Alternative Grid based Strategies.

Another line of attempts to find a discretization of the set $\hat{\Gamma}^{(\alpha)}$ consist in roughly approximating the conformal set by restricting $\hat{\Gamma}^{(\alpha)}(x_{n+1})$ to an arbitrary fine grid of candidate $\hat{\mathcal{Y}}$ i.e. $\bigcup_{z \in \hat{y}}[x_{n+1}^\top \hat{\beta}(z) \pm \hat{Q}_{1-\alpha}(z)]$ [16]. Such approximation did not show any coverage guarantee. To overcome this issue, [6] have proposed a discretization strategy with a more carefully rounding procedure of the observation vectors.

Given an arbitrary finite set $\hat{\mathcal{Y}}$ and any discretization function $\hat{d} : \mathbb{R} \mapsto \hat{\mathcal{Y}}$, define

$$\Gamma^{\alpha, 1} = \{z \in \mathbb{R} : \hat{d}(z) \in [x_{n+1}^\top \hat{\beta}(d(z)) \pm \hat{Q}_{1-\alpha}(\hat{d}(z))]\} \ , \tag{27}$$

$$\Gamma^{\alpha, 2} = \bigcup_{z \in \hat{y}} \hat{d}^{-1}(z) \cap [x_{n+1}^\top \hat{\beta}(z) \pm \hat{Q}_{1-\alpha}(z)] \ . \tag{28}$$

Then [6][Theorem 2] showed that for any exchangeable finite set $\tilde{\mathcal{Y}} = \tilde{\mathcal{Y}}(\mathcal{D}_{n+1})$ and discretization function $\tilde{d} : \mathbb{R} \mapsto \tilde{\mathcal{Y}}$, we have the coverage

$$\mathbb{P}^{n+1}(y_{n+1} \in \Gamma^{\alpha, i}) \geq 1 - \alpha - \mathbb{P}^{n+1}((\hat{\mathcal{Y}}, \hat{d}) \neq (\tilde{\mathcal{Y}}, \tilde{d})) \text{ for } i \in \{1, 2\} \ . \tag{29}$$

A noticeable weakness of this result is that it strongly depends on the relation between the couples $(\hat{\mathcal{Y}}, \hat{d})$ and $(\tilde{\mathcal{Y}}, \tilde{d})$. Equation (29) fails to provide any meaningful informations in many situations e.g. the bound is vacuous anytime $|\hat{\mathcal{Y}}| \neq |\tilde{\mathcal{Y}}|$ or two different discretizations are chosen. Thus, the sets $\Gamma^{\alpha, 1}, \Gamma^{\alpha, 2}$ need a careful choice of the finite grid point $\hat{\mathcal{Y}}$ to be practical. This paper shows how to automatically and efficiently calibrate such set without loss in the coverage guarantee. Our approach provide optimization stopping criterion for each grid point while for arbitrary discretization, one must solve problem (3) at unnecessarily high accuracy. Last but not least, when the loss function is smooth, our approach provide an unprecedented guarantee to contains the full, exact conformal set.

## 6.5 Additional Experiments

### 6.5.1 Sparse Nonlinear Regression

We run experiments on the `Friedman1` regression problem available in `sklearn` where the inputs $X$ are independent features uniformly distributed on the interval $[0, 1]$. The output $z$ is nonlinearly generated using only 5 features.

$$y = 10\sin(\pi X_{:,1}X_{:,2}) + 20(X_{:,3} - 0.5)^2 + 10X_{:,4} + 5X_{:,5} + 0.5\mathcal{N}(0, 1) \ . \tag{30}$$

The results are displayed in Table 3

|          | Oracle | Split | 1e-2  | 1e-4  | 1e-6  | 1e-8  |
|----------|--------|-------|-------|-------|-------|-------|
| `Lasso`  |        |       |       |       |       |       |
| Coverage | 0.91   | 0.88  | 0.93  | 0.89  | 0.89  | 0.89  |
| Length   | 1.50   | 2.320 | 2.272 | 2.011 | 1.956 | 1915  |
| Time     | 0.005  | 0.003 | 0.020 | 0.076 | 0.397 | 3.499 |

Table 3: Computing conformal set for `lasso` regression problem `friedman1` dataset with $n = 506$ observations and $p = 13$ features (resp. $n = 500$ and $p = 50$). We display the coverage, length and time of different methods averaged over 100 randomly left out validation data.

### 6.5.2 Real Data with Large Number of Observations

In this benchmark, we illustrate the performances of the different conformal prediction strategies when the number of observations is large. We use the California housing dataset available in `sklearn`. Results are reported in Table 4.

|                            | Oracle | Split | 1e-2  | 1e-4  | 1e-6  | 1e-8  |
|----------------------------|--------|-------|-------|-------|-------|-------|
| `Smooth Chebychev Approx.` |        |       |       |       |       |       |
| Coverage                   | 0.92   | 0.92  | 0.92  | 0.92  | 0.92  | 0.92  |
| Length                     | 0.014  | 0.014 | 0.014 | 0.014 | 0.014 | 0.014 |
| Time                       | 0.096  | 0.065 | 0.203 | 0.269 | 0.942 | 7.578 |

Table 4: Computing conformal set for `logcosh` regression problem regularized with Ridge penalty on California housing dataset with $n = 20640$ observations and $p = 8$ features. We display the coverage, length and time of different methods averaged over 100 randomly left out validation data.

In this example, both the Splitting and the proposed homotopy method achieves the same performances than the `Oracle` (which use the target $y_{n+1}$ in the model fitting). Due to the large number of observations $n$, the efficiency of the Splitting approach is less affected by its inherent reduction of the sample size. We still note that, a rough approximation of the optimal solution is sufficient to get a good conformal prediction set with homotopy.