[Reviews · NeurIPS 2019]

Reviewer 1



The authors clearly addressed my concerns so I raised my score from 6 to 7. -------------------------------------------------------------------------- The paper is well written and the methodology is potentially useful in this literature. However I have a few major concerns. (1) The exact conformal prediction set may not be an interval but a union of multiple intervals in principle. This is problematic from a statistical and a practical point of view. That is partly the reason why split conformal set (e.g. Lei et al. 2018) or cross-validated conformal set (e.g. Barber et al. 2019) is preferred because they guarantee the set to be an interval despite some efficiency loss. For instance, if the prediction set is [3, 5]U[7, 9], how can practically interpret why 5 and 7 are in the set while 6 is not? Back to the exact conformal prediction set, in most cases we do not need this set other than its left and right endpoints, say C_L and C_U, based on which we can construct a confidence interval [C_L, C_U]. However, much of the efforts in this paper are made on approximating the whole set which appears to be an overkill. For this reason, I do not see why we need to use the homotopy method to approximate the exact conformal set. (2) How do you choose y_min and y_max? Underestimating y_max or overestimating y_min may significantly reduce the coverage. There must be a theoretically guaranteed way to select these two parameters. (3) On the other hand, if they can be chosen, as mentioned in point (1), we only need to find C_L = \inf\{y > y_min: \pi(y) > \alpha\} and C_U = \sup\{y < y_max: \pi(y) > \alpha\}, which seems to be much easier problem than approximating the whole conformal set. Minor points: (1) Equation (15): the right parenthesis in the denominator is missing. (2) Proposition 5 in Supplementary Material: equation (30), G_y(\beta(y, \theta(y))) --> G_y(\beta(y), \theta(y)); line 386, D_y(\theta) --> D_y(theta(y)); line 386, \theta - \hat{\theta}(y) --> \theta(y) - \hat{\theta}(y). References Jing Lei, Max G’Sell, Alessandro Rinaldo, Ryan J Tibshirani, and Larry Wasserman. Distribution-free predictive inference for regression. Journal of the American Statistical Association, 113(523):1094–1111, 2018. Barber, R. F., Candes, E. J., Ramdas, A., & Tibshirani, R. J. (2019). Predictive inference with the jackknife+. arXiv preprint arXiv:1905.02928.

Reviewer 2



The paper addresses the topic of efficient computation of high-confidence prediction sets in the context of regression in conformal prediction. The underlying regression model is the empirical-risk minimizer over linear functions, potentially with a convex regularization on the parameters. This research topic is quite significant, since computation of these prediction sets is notoriously expensive. The approach taken by the authors is interesting and promising, however, the paper lacks at several points. For one, the paper is rather hard to read. Sections 1-2 are well written and give an easy introduction to the topic. From section 3 on (and especially in section 4), however, the flow is broken; the paper digresses towards side remarks and fails to concisely explain its approach. The main algorithm for computation of the prediction sets is never fully stated. At times, notation is unclear (such as using G both for gradient and duality gap, or using \times as a multiplication operation in (20)). Second, it is hard to separate the contribution of this paper from prior work, especially the work on Ridge Regression and Lasso Regression mentioned in the paper ([18] and [14] in the references, respectively). While apparently the warm-start approximate solutions are novel work in this direction, it is not properly explained how this compares to the homotopy approach described especially in [14]. This problem gets even more pronounced when in the experimental section, the proposed method is not properly compared against those methods. For the Ridge Regression, Fig 1 is supposed to give a comparison; however, the figure is very hard to understand, operates on different scales and axes and also gives no implication wrt the computation timing - which is supposed to be the main advantage of the novel method. For the Lasso, a comparison to [14] is completely missing from the experimental section, which arguably should be the closest related work to this paper. (although a comparison to oracle & split methods is provided) A third (minor) shortcoming of this paper is that there are many typos and questionable style formats (almost every paragraph, e.g. very regularly using singular when plural would be appropriate; sometimes headings end with a dot, sometimes not; Statements/Lemmas are differently numbered in the appendix than in the main text) All in all, while the research topic and contributions look promising, the paper doesn't feel mature enough for publication, and it is unclear how it will actually compare empirically against state-of-the-art methods.

Reviewer 3



1. Originality: The technique proposed in the paper to update an approximate solution to certain optimization problems for streaming data is inspired from previous work on homotopy continuation, but is an interesting and new contribution. Combined with its use for efficiently constructing conformal sets, the contribution is novel. The paper clearly indicates prior work and distinguishes itself from the existing methods. 2. Quality: The submission is technically sound. The theoretical claims are supported with proofs and the work represents relatively complete body. 3.Clarity: The paper is clear and well-written for the most part. Please see improvements for further comments. 4. Significance: The paper proposes an efficient method for an emergency problem of interest to construct distribution-free prediction intervals for machine learning problems. The numerical performance of the method indicates promising results compared to an existing method and is likely to draw interest from researchers working on similar problems. The technique used to build the method is additionally also of independent interest and potentially might have other useful applications.

[Author Response · NeurIPS 2019]

1  We would like to thank the reviewers for their constructive feedbacks and we will correct the typos raised and include
2  the suggestions for improving the paper readability accordingly. We would appreciate if the reviewers positively updated
3  their ratings if our responses were satisfactory.

# 1  Reviewer #1

**Full (exact) conformal set vs. split or cross-validated conformal set**  *Full (exact)* conformal prediction set is
important and worth studying since statistical efficiencies are lost both in the model fitting stage and conformity score
rank computation stage in split or cross-validated approach. This is visible in many experiments conducted in previous
papers [14, 15] and confirmed in ours.

**Non-connectedness of the conformal prediction set.**  In practice, we consider the convex hull of the conformal set,
which is always an interval. This was initially suggested in [18, Remark 1]. The lack of interpretability is a real issue,
but it is an intrinsic default of the conformal set which remains in our proposed computation.

**Choice of $[y_{\min}, y_{max}]$.**  We follow the actual practice in the literature [14, Remark 5]. We choose $y_{\min} = y_{(1)}$ and
$y_{\max} = y_{(n)}$. In that case, we have $\mathbb{P}(y_{n+1} \in [y_{\min}, y_{\max}]) \geq 1 - 2/(n+1)$. This implies a loss in the coverage
guarantee of $2/(n+1)$, which is negligible when $n$ is sufficiently large. We did not observe violations.

**Direct fitting of $C_L$ and $C_U$.**  We do not agree that computing the whole path, with homotopy, is not needed for
computing the lower and upper bound (even for Lasso and Ridge). Defining $C_L = \inf\{y > y_{\min} : \pi(y) > \alpha\}$ and
$C_U = \sup\{y < y_{max} : \pi(y) > \alpha\}$, computing $C_L$ and $C_U$ is as hard as computing the exact conformal set. Indeed,
for simplicity assume that the full exact conformal set is an interval. Then, we have

$$[C_L, C_U] = \hat{\Gamma}^{(\alpha)}(x_{n+1}) \cap [y_{\min}, y_{max}] \ . \tag{1}$$

Unless an explicit and simple enough formula for $\pi(y)$ (i.e. $\hat{\beta}(y)$) is available, such computation are intractable and
was, so far, limited to class of regression problem where the entire solution path for $y$ can be computed exactly. Our
contribution, based on approximated solution and convex hull, can be interpreted as a direct estimation of $C_L$ and $C_U$.

# 2  Reviewer #2

**Paper readability.**  Thanks, we will fix the notation issues and put some remarks and details in the appendix to ease
the reading flow. We will also summarize the proposed algorithm in a direct pseudo-code. While important, we believe
that the presentation issues pointed out here can be properly corrected in the camera-ready version without changing the
main technical body of the work.

**Contributions compared to prior work.**  As discussed in line 99, the papers [18] and [14] are restricted to Ridge
and Lasso where *explicit* solution are available as a piece of linear function of $y$. In that case, approximation is no
longer needed since the exact conformal set can be efficiently computed (with *only* a single model fitting). Our main
contribution (which is not limited to linear model) is *not* to provide more computationally efficient method than existing
ones but to provide an easily computable conformal set, based on approximated solution, when the exact set *cannot* be
computed (we have presented the `logcosh` and `Linex` loss function as examples).

Figure 1 merely illustrates the trade-off between the statistical efficiency (length of the interval) and optimization error
$\epsilon$ (in case of Ridge where the exact set can be computed). We will clarify this; including suggestions of Reviewer #4.

Contrarily to previous methods, our approach provide a simple, general and *unified* framework for computing full
conformal set under mild assumptions on the loss function and *any* convex regularization $\Omega$, along with a transparent
complexity analysis (which is still unknown for the exact homotopy in Lasso; note in general that *worst case* complexity
of *exact* homotopy can be exponential in the dimension of the underlying optimization problem (Gartner et al., 2012)).
Thus, we generalize [18] and [14] to a much wider class of machine learning problems (See answer to Reviewer #1).

# 3  Reviewer #4

**Improvement of the numerical experiments.**  Thanks, we highly appreciate your critiques. The suggested experi-
ments will be investigated and added in our paper. By doing a quick check, in the Lasso case, we observe consistent
results when the coverage level $\alpha$ varies. This is an important point and we will perform a more complete experiments
and properly report the results and our understanding.

[Meta-Review · NeurIPS 2019]

The authors study fast approximations to the "full" conformal prediction set using homotype-type algorithms for penalized convex problems. There are some interesting ideas here, and the reviewers were mostly positive, but pointed out a number of relevant critiques. All in all, I'm leaning towards the favorable, however I have two concerns that the authors really must address in a camera-ready version. First the experiments need to improve. Claiming to not run comparisons against efficient ridge [18] or efficient lasso [11] because these problems are already solved is not a good excuse. It is of course instructive to also give examples of your algorithm where it is not ideal so that we can see how much it loses to exact methods and therefore gain an understanding of what's happening with the approximations. Plus the experiments need to improve in general, as pointed out by Reviewer 4. Second, the paper's explanations are a bit confusing to me in places. For example, even the basic explanation of conformal prediction needs to improve. I understand this methodology well so I understood what the authors are going for, but a reader not familiar with conformal would be lost. Nowhere is Rank() defined (rank with respect to **what** set? of course this is critical). And references to [5] and [13] are completely unecessary and need to be removed. It should be explained **from first principles** what is going on here, not references books/papers on exchangeability or testing.